# Oleoylethanolamide Treatment Modulates Both Neuroinflammation and Microgliosis, and Prevents Massive Leukocyte Infiltration to the Cerebellum in a Mouse Model of Neuronal Degeneration

**DOI:** 10.3390/ijms24119691

**Published:** 2023-06-02

**Authors:** Ester Pérez-Martín, Laura Pérez-Revuelta, Cristina Barahona-López, David Pérez-Boyero, José R. Alonso, David Díaz, Eduardo Weruaga

**Affiliations:** 1Laboratory of Neuronal Plasticity and Neurorepair, Institute of Neuroscience of Castile and Leon (INCyL), Universidad de Salamanca, 37007 Salamanca, Spain; esterpm@usal.es (E.P.-M.);; 2Institute of Biomedical Research of Salamanca (IBSAL), 37007 Salamanca, Spain

**Keywords:** endocannabinoids, microglia, neurodegeneration, neuroinflammation, neurotherapeutics, oleoylethanolamide (OEA), PCD mouse, Purkinje cells

## Abstract

Neurodegenerative diseases involve an exacerbated neuroinflammatory response led by microglia that triggers cytokine storm and leukocyte infiltration into the brain. PPARα agonists partially dampen this neuroinflammation in some models of brain insult, but neuronal loss was not the triggering cause in any of them. This study examines the anti-inflammatory and immunomodulatory properties of the PPARα agonist oleoylethanolamide (OEA) in the Purkinje Cell Degeneration (PCD) mouse, which exhibits striking neuroinflammation caused by aggressive loss of cerebellar Purkinje neurons. Using real-time quantitative polymerase chain reaction and immunostaining, we quantified changes in pro- and anti-inflammatory markers, microglial density and marker-based phenotype, and overall leukocyte recruitment at different time points after OEA administration. OEA was found to modulate cerebellar neuroinflammation by increasing the gene expression of proinflammatory mediators at the onset of neurodegeneration and decreasing it over time. OEA also enhanced the expression of anti-inflammatory and neuroprotective factors and the *Pparα* gene. Regarding microgliosis, OEA reduced microglial density—especially in regions where it is preferentially located in PCD mice—and shifted the microglial phenotype towards an anti-inflammatory state. Finally, OEA prevented massive leukocyte infiltration into the cerebellum. Overall, our findings suggest that OEA may change the environment to protect neurons from degeneration caused by exacerbated inflammation.

## 1. Introduction

Neuroinflammation is a complex and dynamic innate immune process that occurs within the central nervous system (CNS), orchestrated by different glial cells, cytokines, and chemokines, among others, which initially plays a part as the first line of defense when the CNS encounters different insults [1,2]. As resident macrophages in the CNS, microglial cells play a key role in mediating these neuroinflammatory processes [1]. In an unstimulated state, resting branch-shaped microglia surveys the microenvironment and contribute to the maintenance of brain homeostasis by supporting neuronal survival, physiological cell death, and synaptogenesis [3]. Upon activation by pathological stimuli such as infection, brain trauma, stroke, and neuronal degeneration [3,4], microglia undergoes a rapid change in their morphology (from highly ramified to an amoeboid shape) [5] and distribution, and initiate a series of molecular events leading to the production of pro- or anti-inflammatory mediators in brain parenchyma [1,3,4]. Traditionally, microglia has been simplistically classified in a dichotomous manner: M1 and M2 (i.e., pro- and anti-inflammatory states, respectively) based on the injury and the stimulus inducing activation [4,6]. However, several studies examining microglial transcriptomics and proteomics have revealed that such activation consists of a broad spectrum ranging from pro- to anti-inflammatory states, and it is characterized by variable and complex expression of different markers, cytokine production, and functions [4,6,7]. Moreover, the release of cytokines and chemokines by microglial cells promotes the recruitment of peripheral leukocytes to the brain, further contributing to the neuroinflammatory state of the CNS [2,8,9,10]. Thus, microglia can play a double-edged role since, initially, its activation aims to protect the CNS, and an exacerbated and chronic activation leads to a maintained neuroinflammatory state that becomes highly detrimental to neurons [2]. Indeed, emerging evidence has suggested that this dysregulated CNS innate immunity sustained over time has been implicated in the onset and progression of various neurological and neurodegenerative disorders, such as Alzheimer’s and Parkinson’s disease [1]. Therefore, the chronic activation of microglia has become a significant focus of research for the development of therapeutic strategies for treating neurodegenerative disorders.

The crucial role of non-classical endocannabinoids in the intrinsic response to neuroinflammation is well documented (for review, [11]). In particular, peroxisomal proliferator-activated receptor alpha (PPARα) plays an important role by regulating inflammatory processes and has emerged as a common target in diseases involving inflammation [12,13]. In this regard, some studies have reported that the endocannabinoid oleoylethanolamide (OEA), a ligand of PPARα [14], mediates anti-inflammatory effects by decreasing inflammatory cytokine levels, increasing neuroprotective factors, and modulating glial activation [15,16,17,18,19,20,21,22,23,24,25,26]. These effects have been observed in different animal models of neuroinflammation with diverse etiologies: LPS injection [16,17,18], brain ischemia [19,21,25], stress of diverse etiology [20,24,26], and excessive ethanol consumption [22,23]. Despite these promising findings, to the best of our knowledge, no study to date has addressed whether OEA treatment might play an immunomodulatory effect on neuroinflammation triggered by primary neuronal loss, such as that observed in neurodegenerative diseases. Indeed, in a recent study conducted by our research group, we demonstrated that the same dose and timing of OEA administration used in the present work (10 mg/kg; at P12) exerted neuroprotection both at the histological and behavioral levels through PPARα in the same model of cerebellar neurodegeneration employed here, the Purkinje Cell Degeneration (PCD) mutant mouse [27]. In particular, OEA delayed morphological changes and death of the main affected neuronal population affected, and ameliorated the motor, cognitive, and social neurobehavioral defects in PCD mice [27]. This animal is characterized by early onset cerebellar atrophy caused by a mutation in the *Ccp1/Agtpbp1/Nna1* gene (MGI/NCBI IDs: 2159437/67269 http://www.informatics.jax.org/marker/MGI:2159437 (accessed on 22 January 2023)) and a consequent lack of cytosolic carboxypeptidase 1 enzymatic activity [28,29,30,31,32,33,34,35]. Moreover, the PCD mutant mouse displays pathophysiological and clinical manifestations similar to Childhood-Onset Neurodegeneration with Cerebellar Atrophy (CONDCA) observed in humans with a monogenic biallelic mutation in *CCP1* [36,37,38,39]. In the PCD mouse model, the mutation causes the drastic death of Purkinje cells from the third week of postnatal development [29,32,40]. Specifically, neuronal degeneration begins as an initial predegenerative phase, from postnatal day 15 (P15) to P18, during which nuclear, cytological, and morphological changes are observed in Purkinje cells [8,41,42,43,44]. Then, it progresses to a degenerative stage per se from P18 onwards, in which Purkinje cells rapidly die with only a few remaining in the lobe X of the cerebellar vermis at P40 [28,31,42,43,44,45,46,47]. This neuronal degeneration leads to severe cerebellar microgliosis and leukocyte recruitment to cerebellar regions where neurodegeneration is most aggressive (i.e., lobe I to IX and the outermost layers of the cerebellar cortex) [8,10,40]. In sum, the PCD mouse is an extremely suitable model for exploring therapeutic strategies for the treatment of severe neurodegenerative disorders involving neuroinflammation, in addition to having a direct translational application for treating human diseases.

The present study aims to assess the potential therapeutic effect of the endocannabinoid OEA on neuroinflammation and microgliosis caused by neuronal degeneration in the PCD mutant mouse. To this end, we employed the dose and timing of OEA (10 mg/kg; at P12) previously described as the most effective treatment in PCD mice [27] and we performed different molecular and histological analyses throughout the neuronal degenerative process (see below). To begin with, we first analyzed changes in the gene expression of different proinflammatory, anti-inflammatory, and neuroprotective factors in the cerebellum after OEA treatment in the short, medium, and long term. Secondly, we assessed the changes in the density, distribution, and marker-based phenotype of microglial cells at P30, when Purkinje cell degeneration in PCD mice is at its peak. Finally, we also examined the overall density and distribution of leukocytes in the cerebellar parenchyma at P30. The experimental design and timeline are shown in Figure below. It is important to highlight that both wild-type (WT) and untreated PCD mice were included as control groups, since the aim of the present work was not only to examine the effects of OEA treatment under a pathological condition, but also to establish a comparison with the baseline of healthy individuals.

## 2. Results

### 2.1. OEA-Modulated Cerebellar Gene Expression of Proinflammatory and Anti-Inflammatory/Neuroprotective Factors

We evaluated the gene expression of different inflammatory markers (*Cox2*, *Ifnγ*, *Il1β*, *Il6, iNos*, *Nfκb*, and *Tnfα*) and neuroprotective factors (*Bdnf*, *Gap43*, and *Map2*) related to the mechanism of action of OEA and the receptor through which OEA exerts its main actions (*Pparα*) in the short- (3 and 24 h after OEA administration at P12), medium- (P20), and long term (P30) after OEA administration (10 mg/kg, i.p., at P12). Differences in mRNA levels of these markers were found in the cerebellum at the three time points considered (Figure 1). To facilitate the understanding of these results, each of the time points analyzed will be presented separately below.

In the short term, OEA effects were exclusively performed in unharmed WT mice (see Section 4), since at this age (P12-13) PCD animals do not exhibit any differences compared to their WT littermates and neurodegeneration has not yet started [32,41,43]. Effects of OEA on mRNA levels at 3 and/or 24 h after treatment were detected for the inflammatory markers *Ifnγ*, *Il1β*, *Il6*, and *Tnfα*; for the neuroprotective markers *Bdnf*, *Gap43*, *Map2* and its receptor *Pparα* (*p Cox2* = 0.459; *p Ifnγ* = 0.011; *p Il1β* = 0.016; *p Il6* = 0.025; *p iNos* = 0.158; *p NfκB* = 0.272; *p Tnfα* = 0.010; *p Bdnf* = 0.022; *p Gap43* = 0.015; *p Map2* = 0.003, and *p Pparα* = 0.003; specific *p*-values for multiple comparisons in Table 1; Figure 1c,d). In general, the mRNA expression of these inflammatory markers decreased at 3 and/or 24 h after OEA administration, except for *Il6* whose mRNA expression level increased 3 h after OEA was applied but decreased to basal (untreated WT) levels 24 h later (Table 1; Figure 1c,d). By contrast, the gene expression of neuroprotective factors and the *Pparα* receptor increased at both 3 and 24 h after OEA treatment for all the genes analyzed (Table 1; Figure 1c,d). These results, in WT animals, indicated that OEA has an anti-inflammatory and neuroprotective effect, even in an unharmed brain.

Concerning the medium-term analysis at P20, when neurodegeneration of PCD mice had just begun to lead to an acute neuroinflammatory state [40], differences were detected among WT, untreated PCD, and OEA-treated PCD animals in the gene expression of both inflammatory and neuroprotective factors, and in *Pparα* receptor (*p Cox2* = 0.006; *p Ifnγ* = 0.097; *p Il1β* = 0.002; *p Il6* = 0.0003; *p iNos* = 0.018; *p NfκB* = 0.006; *p Tnfα* = 0.010; *p Bdnf* = 0.003; *p Gap43* = 0.008; *p Map2* = 0.024; and *p Pparα* = 0.048; specific *p*-values in Table 1; Figure 1e,f). First, we found a striking increase in *Tnfα* expression in PCD animals compared to their WT counterpart, whereas the expression of the rest of the proinflammatory factors remained unchanged or even decreased as observed in *iNos* expression (Table 1; Figure 1e,f). Moreover, a decrease in mRNA levels of *Gap43* and *Pparα* expression was also detected in PCD mice compared to WT animals at this time point (Table 1; Figure 1e,f). Interestingly, OEA administration increased the mRNA levels of half of the factors analyzed at P20 compared to the baseline: *Cox2*, *Ifnγ*, *Il1β*, *Il6*, and *Tnfα* (Table 1; Figure 1e,f). Furthermore, OEA-treated PCD mice showed a higher level of gene expression of some neuroprotective factors, such as *Bdnf*, *Gap43*, and *Map2*, compared to untreated PCD mice, and presented levels similar to that of the wild type (Table 1; Figure 1e,f; ref. [47]).

Regarding long-term analysis at P30, when Purkinje cell degeneration was at its peak and chronic neuroinflammation had set in [40], differences in mRNA levels of inflammatory and neuroprotective factors, and *Pparα* receptor, were also detected among WT, untreated PCD, and OEA-treated PCD mice (*p Cox2* = 0.587; *p Ifnγ* = 0.022; *p Il1β* = 0.008; *p Il6* = 0.010; *p iNos* = 0.050; *p NfκB* = 0.548; *p Tnfα* = 0.004; *p Bdnf* = 0.008; *p Gap43* = 0.082; *p Map2* = 0.050; and *p Pparα* = 0.008; specific *p*-values for multiple comparisons in Table 1; Figure 1g,h). First, untreated PCD mice showed increased mRNA levels of the following proinflammatory factors: *Ifnγ*, *Il1β*, *Il6*, and *Tnfα*. Conversely, a decrease in the neuroprotective *Bdnf* and *Pparα* was detected compared to WT levels (Table 1; Figure 1g,h). Therefore, these results indicate that there is an increase in the neuroinflammatory state at this time point in the PCD mutant mouse which is consistent with the neurodegenerative process occurring at the same time. Nevertheless, in the long term after OEA administration, mRNA levels of the proinflammatory markers decreased either to those observed in WT mice for *Ifnγ*, *Il1β*, and *Il6* (Table 1; Figure 1g,h), or to an intermediate level between WT and untreated PCD for *Tnfα* (Table 1; Figure 1g,h). By contrast, *Bdnf*, *Map2*, and *Pparα* mRNA levels were increased in OEA-treated PCD mice compared to those of untreated PCD mice, reaching WT levels or even higher (Table 1; Figure 1g,h; ref. [47]).

All these medium- and long-term results indicate that OEA may exert a dual effect on the degenerating cerebellum of PCD mice over time. At the onset of cerebellar degeneration, OEA increased gene expression of almost all genes evaluated, whereas in the long term it modulated cerebellar neuroinflammation by decreasing gene expression of proinflammatory markers and increasing that of neuroprotective factors [47].

Finally, we analyzed the effect of OEA on hemi-cerebellum weight and the ratio to total body weight (Appendix A). Differences in left hemi-cerebellum weight were found between WT and PCD mice, regardless of the treatment, in the medium- and long term (Appendix A). Moreover, OEA-treated PCD mice showed an overall increase in the ratio of hemi-cerebellum weight to total body weight compared to WT mice in the long term, suggesting a potential reduction in cerebellar atrophy compared to untreated PCD mice (Appendix A).

### 2.2. OEA Treatment Decreased the Density of Iba1-Positive Cells and Partially Modified Their Distribution in the Cerebellum of PCD Mice

Previous works have qualitatively described the microgliosis occurring in the cerebellum of the PCD mouse [8,40]. However, to the best of our knowledge, no studies have addressed a quantitative analysis of changes in microglial density and distribution in this mutant background. Thus, to quantitatively evaluate the effect of OEA treatment on cerebellar microgliosis of PCD mice, we examined microglial density both in total vermis (as a whole) and in this same structure sorted by lobes at P30 (Figure 2) and layers of the cerebellar cortex, as neurodegeneration is not uniform across the cerebellum of PCD mice (Figure 3) [32,40,45].

Regarding total microglial density, an increase in the number of Iba1-positive cells per mm^2^ was found throughout the entire cerebellar vermis of untreated PCD mice at P30 compared to the wild type (*p* ANOVA < 0.001; specific *p*-values for multiple comparisons in Table 2; Figure 2a,b). This increase was partially prevented when PCD animals were treated with OEA, showing values closer to those of WT mice (WT: 218.63 ± 12.23 cells/mm^2^, PCD: 719.95 ± 24.79 cells/mm^2^, PCD + OEA: 519.83 ± 16.31 cells/mm^2^; Table 2; Figure 2b).

Given that cerebellar neurodegeneration does not occur homogeneously in PCD mice, and that lobe X is more resistant than the rest of the cerebellar lobes [32,45], we analyzed the distribution of microglial cells by cerebellar lobes. Thus, we evaluated changes in Iba1-positive cell density in lobes (L) I to IX vs. lobe X (LI-IX and LX, respectively) within each experimental group (Figure 2c), and among the different experimental groups (Figure 2d). Differences in Iba1-positive cell density were detected between LI-IX and LX in both groups of PCD mice (*p* < 0.001 for both experimental groups), with the density being higher in LI-IX than in LX (PCD: 798.46 ± 31.23 cells/mm^2^ vs. 483.54 ± 19.17 cells/mm^2^, PCD + OEA: 557.82 ± 15.31 cells/mm^2^ vs. 405.87 ± 27.86 cells/mm^2^, LI-IX and LX, respectively; Figure 2). However, no statistically significant differences were found between the lobes of WT mice (*p* = 0.978; 214.75 ± 7.59 cells/mm^2^ vs. 215.67 ± 31.53 cells/mm^2^, LI-IX and LX, respectively; Figure 2c). These results indicated that PCD microglial cells were not equally distributed throughout the entire cerebellum, showing a tropism towards LI-IX, where neuronal degeneration is more aggressive. Furthermore, differences among the three experimental groups were found in microglial cell density from LI-IX, with the values obtained from PCD OEA-treated animals being halfway between those obtained for both WT and untreated PCD mice (*p* ANOVA < 0.001; specific *p*-values in Table 2; Figure 2d). By contrast, only differences between the WT and the two PCD groups were detected in LX regardless of OEA treatment (Table 2; Figure 2d). Thus, these results indicate that the observed changes in total microglial density (Figure 2b) are determined mainly by the microglial density of LI-IX (Figure 2d).

Finally, since the *pcd* mutation affects mainly Purkinje cells in the cerebellar cortex [29,32,40], we also chose to separately analyze the distribution of microglia within the different layers of the cerebellar cortex: molecular layer (ML) + Purkinje cell layer (PCL) versus granular layer (GL). Thus, we quantified the microglial density in ML and PCL together, where Purkinje cell dendritic arbors and somata are located, and separately in the GL (Figure 3). Differences in Iba1-positive cells per mm^2^ were detected between the two regions in the entire cerebellum in the three experimental groups (*p* WT= 0.025; *p* PCD < 0.001; *p* PCD + OEA < 0.001; Figure 3b). Interestingly, in contrast to WT mice, in both PCD groups a higher density of Iba1-positive cells was found in ML + PCL compared to GL (WT: 196.46 ± 6.76 cells/mm^2^ vs. 237.43 ± 13.22 cells/mm^2^; PCD: 1038.95 ± 38.09 cells/mm^2^ vs. 447.30 ± 33.74 cells/mm^2^; and PCD + OEA: 787.36 ± 21.53 cells/mm^2^ vs. 248.78 ± 12.25 cells/mm^2^, ML + PCL and GL, respectively; Figure 3b). These differences in microglial distribution were maintained separately in LI-IX (*p* WT = 0.009, WT: 193.03 ± 4.99 cells/mm^2^ vs. 241.19 ± 12.97 cells/mm^2^; *p* PCD < 0.001, PCD: 1186.41 ± 43.15 cells/mm^2^ vs. 475.84 ± 46.49 cells/mm^2^; *p* PCD + OEA < 0.001, PCD + OEA: 880.97 ± 18.85 cells/mm^2^ vs. 248.76 ± 10.53 cells/mm^2^, ML + PCL and GL, respectively; Figure 3c) and in LX (*p* WT = 0.489, WT: 201.72 ± 12.11 cells/mm^2^ vs. 218.12 ± 19.11 cells/mm^2^; *p* PCD < 0.001, PCD: 587.90 ± 24.32 cells/mm^2^ vs. 361.55 ± 21.94 cells/mm^2^; *p* PCD + OEA < 0.001, PCD + OEA: 506.52 ± 37.72 cells/mm^2^ vs. 248.85 ± 19.88 cells/mm^2^, ML + PCL and GL, respectively; Figure 3d). Although it seemed that microglial cells behaved similarly in terms of distribution in all PCD animals, differences were detected among the three experimental groups when the microglial density (separated both per lobe and layer) was analyzed (*p* ANOVA < 0.001 for all; specific *p*-values of multiple comparisons in Table 2; Figure 3e–g). Overall, in the ML + PCL analysis, a partial effect of the OEA treatment was observed, with the values obtained from the OEA-treated PCD mutant being midway between those of untreated PCD and WT mice (Table 2; Figure 3e–g). However, in the GL analysis, a full effect was detected after OEA treatment, as no statistically significant differences were observed between OEA-treated PCD and WT mice in all comparisons made (Table 2; Figure 3e–g).

### 2.3. OEA Administration Modified Microglial Population towards an Anti-Inflammatory Phenotype by Increasing the Percentage of CD206/Iba1-Positive Cells

Considering that in terms of microgliosis, both the quantity and the microglial phenotype are important [48], we decided to analyze in the cerebellar vermis, at P30, three of the most common microglial markers within the Iba1-positive cell population: CD45^low^ (reactive microglial marker [10,49], CD16/32 (proinflammatory marker), and CD206 (anti-inflammatory/tissue repair marker [48], as shown in Figure 4 and in Appendix A, where a representation and a detailed magnification of the aforementioned immunofluorescence combinations are provided, respectively. According to the prism plots, representing the individual percentage of the different microglial populations analyzed in each vertex for each experimental group, the WT tissue had a noticeably lower percentage of all microglial populations compared to that of both PCD groups, untreated and OEA-treated PCD mice (Figure 5a). Moreover, differences were detected upon comparing the prisms of the PCD groups in the percentage of the anti-inflammatory CD206 marker microglial population, whereas CD45 and CD16/32 markers seemed to not be affected by OEA treatment (Figure 5a). To facilitate the understanding of the quantitative analyses, each microglial marker will be explained separately below. Specific *p*-values of multiple comparisons are summarized in Table 3.

Regarding the microglial population positive for CD45 (reactive microglial marker; Figure 4 and Appendix A), we detected an increase in the total percentage of double-stained CD45/Iba1-positive cells in both PCD groups compared to the wild type, although no differences derived from OEA treatment were found (WT: 11.77 ± 1.18%, PCD: 37.20 ± 1.63%, PCD + OEA: 37.08 ± 1.41%; *p* ANOVA < 0.001, specific *p*-values for multiple comparisons in Table 3; Figure 5b). This pattern was also observed when comparing the percentages obtained for each lobe among the three experimental groups (LI-IX, WT: 13.40 ± 1.61%, PCD: 35.86 ± 1.14%, PCD + OEA: 38.01 ± 1.62%; LX, WT: 7.82 ± 0.79%, PCD: 41.24 ± 3.38%, PCD + OEA: 34.29 ± 2.37%; Table 3; Figure 5c). Moreover, when comparing the percentage of double-stained CD45/Iba1-positive cells per layer, we found that OEA prevented the peak observed in the GL of the untreated PCD mice, with OEA-treated values being similar to those of the WT mice (Table 3; Figure 5d). However, no OEA effect was detected in ML + PCL, in which PCD groups showed a higher number of double CD45/Iba1-positive cells than WT mice (Table 3; Figure 5d). Finally, we did not detect differences in the distribution of double CD45/Iba1-positive cells throughout the different lobes within the experimental groups (*p* PCD = 0.158; *p* PCD + OEA= 0.221; Figure 5e), except for WT mice, whose LX showed a slight decrease in the percentage of this microglial population (*p* WT= 0.015; 13.40 ± 1.61% vs. 7.82 ± 0.798%; Figure 5e). By contrast, quantitative differences were found between ML + PCL and GL in all the experimental groups (*p* WT = 0.002; *p* PCD < 0.001; *p* PCD + OEA < 0.001; Figure 5f). Interestingly, while the percentage of double CD45/Iba1-positive cells in WT animals was higher in the deeper layers of the cerebellar cortex (i.e., GL), the opposite was found in both PCD groups, which presented a higher percentage of stained cells in the outermost layers (i.e., ML + PCL) where primary degeneration takes place (WT: 2.71 ± 1.02% vs. 9.05 ± 0.88%; PCD: 45.00 ± 1.87% vs. 18.51 ± 2.08%; and PCD + OEA: 46.76 ± 1.78% vs. 9.96 ± 1.03%, ML + PCL vs. GL, respectively; Figure 5f).

Related to the microglial population positive for CD16/32 (proinflammatory marker; Figure 4 and Appendix A), an increase in the total percentage of double CD16/32/Iba1-positive cells was also detected in both PCD groups compared to the WT one (WT: 0.26 ± 0.11%, PCD: 15.42 ± 2.95%, PCD + OEA: 16.15 ± 1.53%; *p* ANOVA < 0.001; specific *p*-values for multiple comparisons in Table 3; Figure 5g). The same results were obtained in the specific analyses of each lobe (LI-IX, *p* ANOVA < 0.001; WT: 0.29 ± 0.15%, PCD: 18.27 ± 3.55%, PCD + OEA: 20.20 ± 2.01%; LX, *p* ANOVA 0.019; WT: 0.20 ± 0.20%, PCD: 7.09 ± 2.09%, PCD + OEA: 4.00 ± 0.85%; Table 3; Figure 5h) and each layer (ML + PCL, *p* ANOVA < 0.001; WT: 0.06 ± 0.06%, PCD: 16.74 ± 3.27%, PCD + OEA: 17.36 ± 1.87%; GL, *p* ANOVA < 0.001; WT: 0.42 ± 0.21%, PCD: 10.66 ± 2.09%, PCD + OEA: 18.51 ± 2.08%; Table 3; Figure 5i). For all these analyses, the PCD groups always presented a higher percentage of CD16/32/Iba1-double-positive cells compared to WT mice. By contrast, we also detected that OEA treatment affected LX by slightly decreasing the density of CD16/32/Iba1-positive cells compared to untreated PCD mice (Table 3; Figure 5h). Next, when comparing differences between LI-IX and LX within the experimental groups, a decrease in the percentage of CD16/32/Iba1-positive cells was observed in the LX of both PCD groups, which is a region more resistant to neurodegeneration and where neuronal damage is less aggressive than in the rest of lobes (*p* WT = 0.741; *p* PCD = 0.019; *p* PCD + OEA < 0.001; Figure 5j). Finally, when we analyzed the differences between ML + PCL and GL within the experimental groups, we only detected an increase in the outermost layers of OEA-treated mice, whereas similar density values were observed in both layers in the WT and untreated PCD mice (*p* WT = 0.148; *p* PCD < 0.144; *p* PCD + OEA < 0.006; Figure 5k).

Finally, concerning the microglial population labeled for detecting CD206 (anti-inflammatory/tissue repair marker; Figure 4 and Appendix A), an increase in the total percentage of double CD206/Iba1-positive cells in the cerebellar cortex was observed for the OEA-treated PCD mice as compared to WT and untreated PCD mice (WT: 1.043 ± 0.45%, PCD: 25.05 ± 2.41%, PCD + OEA: 37.45 ± 3.40%; *p* ANOVA < 0.001; specific *p*-values in Table 3; Figure 5l). A similar result was observed in the per-lobe analysis of LI-IX (*p* ANOVA < 0.001; WT: 0.78 ± 0.36%, PCD: 24.78 ± 2.29%, PCD + OEA: 38.93 ± 3.28%; Table 3; Figure 5m). This pattern was also found in the per-layer analysis of ML + PCL (*p* ANOVA < 0.001; WT: 1.32 ± 0.59%, PCD: 32.92 ± 3.26%; PCD + OEA: 47.67 ± 3.77%; Table 3; Figure 5n) and partially in GL (*p* ANOVA = 0.046; WT: 0.69 ± 0.30%, PCD: 2.99 ± 0.48%, PCD + OEA: 4.79 ± 1.48%; Table 3; Figure 5n). Moreover, no differences in the percentage of this microglial population were found between LI-IX and LX in any of the experimental groups (*p* WT = 0.287; *p* PCD < 0.743; *p* PCD + OEA < 0.001; Figure 5o). However, a higher number of double CD206/Iba1-positive cells was observed in the outermost layers (ML + PCL) of the cerebellar cortex of both PCD groups, as previously observed for the other markers analyzed (*p* WT= 0.368; *p* PCD < 0.001; *p* PCD + OEA < 0.001; Figure 5p).

Overall, considering the last two aforementioned sets of results associated with the effect of OEA on the density and marker-based phenotype of microglial cells (Figure 4 and Figure 5), a decrease in the total density of Iba1-positive cells and a modulation towards an anti-inflammatory state in these cells were observed in PCD animals treated with OEA (Figure 2, Figure 3, Figure 4 and Figure 5).

### 2.4. OEA Treatment Prevented Leukocyte Infiltration towards the Cerebellar Parenchyma

Finally, based on recent work reporting increased leukocyte recruitment in PCD mice due to cerebellar degeneration [10], we explored whether OEA treatment modifies the leukocyte infiltration into cerebellar parenchyma (Figure 6). To this end, we analyzed the overall leukocyte density in the cerebellar vermis of WT, untreated PCD, and OEA-treated PCD mice at P30. Leukocytes were identified as CD45^high^-positive cells with a rounded or rod-like morphology, as previously described [10,50]. Leukocytes were found in all cerebellar lobes and layers in all three experimental groups (Figure 6a). As low values were obtained for each section, we only analyzed total leukocyte density and distribution per lobes (LI-IX and LX) and not per layer. The results of the quantitative analysis showed an increase in total leukocyte density in untreated PCD mice when compared to their WT counterpart (*p* ANOVA < 0.001; specific *p*-values for multiple comparisons in Table 4; Figure 6b). This increase was prevented in OEA-treated animals whose density values were similar to those of WT mice (WT: 5.72 ± 0.86 cells/mm^2^, PCD: 24.80 ± 1.56 cells/mm^2^, PCD + OEA: 8.27 ± 0.80 cells/mm^2^; Table 4; Figure 6b).

Additionally, similar results to those previously mentioned were obtained when differences among the three experimental groups were analyzed separately per lobe (*p* LI-IX < 0.001; *p* LX = 0.001; Table 4; Figure 6c). Finally, we evaluated possible differences in the distribution of leukocytes throughout the cerebellar lobes inside each experimental group. The results of the quantitative analysis showed no statistically significant differences in the overall leukocyte density between LI-IX and LX in any of the experimental groups (*p* WT = 0.067, WT: 4.66 ± 0.76 cells/mm^2^ vs. 8.92 ± 1.45 cells/mm^2^; *p* PCD = 0.073, PCD: 26.57 ± 2.42 cells/mm^2^ vs. 19.47 ± 2.67 cells/mm^2^; and *p* PCD + OEA = 0.591, PCD + OEA: 8.54 ± 1.22 cells/mm^2^ vs. 7.48 ± 1.47 cells/mm^2^, LI-IX and LX, respectively; Figure 6d).

## 3. Discussion

Emerging evidence has shown that PPARα agonists counteract inflammatory responses and have neuroprotective properties in different animal models of brain damage involving neuroinflammation (for review, [11]). In fact, in a recent study conducted by our group, we found that OEA exerts neuroprotection through PPARα in the cerebellum of PCD mutant mice by delaying morphological alterations and death of Purkinje neurons and improving motor, cognitive, and social functions, which were impaired in this model [27]. Moreover, Purkinje cell loss in PCD mice triggers severe gliosis, mainly affecting microglia, characterized by strong morphological changes and enhanced glial proliferation, as well as the release of proinflammatory mediators [8,40]. Given the observed effects of endocannabinoids in modulating neuroinflammation, in the present work we assessed the anti-inflammatory and immunomodulatory properties of the most effective treatment of OEA in PCD mice [27] and its influence on neuroinflammation induced by primary neuronal loss. Since few studies have considered conducting a global quantitative analysis of PCD mouse neuroinflammation, we will first discuss the findings obtained in the mutant model in comparison with WT animals, and then the effects observed after OEA administration.

Regarding the inflammatory state of the PCD cerebellum at P20, when Purkinje cell death has just started [32,39,42], most of the proinflammatory genes analyzed showed no change in their gene expression compared to the baseline. This is consistent with the microglial reaction of PCD mice at this time point, where although present, it is not yet greatly exacerbated [40]. Interestingly, a marked increase in *Tnfα* gene expression was found at this early stage of neurodegeneration, as reported in other neurodegenerative diseases such as Alzheimer’s and Parkinson’s disease [51,52,53]. This result would support the putative role of TNFα as an early onset biomarker of neurodegeneration, as proposed by other authors [51,53]. In contrast, at P30, when most Purkinje cells have degenerated and gliosis is quite evident, an upregulation was detected in the gene expression of the proinflammatory mediators *Ifnγ*, *Il1β*, *Il6*, and *Tnfα* (Figure 7), which is consistent with previous findings [40]. These proinflammatory mediators produced by microglia and macrophages promote neurotoxicity and could also contribute to the aggressive neurodegeneration of Purkinje cells, as occurs in other neurodegenerative disorders such as Alzheimer’s disease [3]. Interestingly, no differences were detected in the gene expression of other proinflammatory molecules or enzymes such as *Cox2* and *Nfκb* in PCD mice. In this regard, although increased activation of both factors has been reported in some brain injury models [54], this response seems to be a rare occurrence in other diseases involving neuroinflammation. Furthermore, some authors suggest the existence of different inflammatory profiles and signaling cascades, as well as the activation of specific inflammasomes, depending on the type of brain damage [55].

Concerning the neuroprotective factors, a downregulation in *Bdnf* gene expression was observed in PCD mice as compared to age-matched WT animals (Figure 7). This is in line with previous work reporting decreased *Bdnf* levels/expression in a broad spectrum of neurological diseases, including PCD mice but at a much older age [56]. The specific cause of this downregulation is unclear; however, it is known that proinflammatory cytokines, especially IL1β, can cause downregulation of *Bdnf* expression [57]. As these cytokines are upregulated in PCD mice at this time point, a decrease in *Bdnf* expression might be expected. In contrast, no changes in *Gap43* and *Map2* gene expression were found in the PCD cerebellum at P30. Indeed, given that the expression of these molecules is mainly localized in cerebellar granule cells [58,59], no differences are expected in PCD mice at this time point when the main neurons affected are Purkinje cells but not granule cells [29,32,44]. Finally, we also analyzed the expression of *Pparα*, as it is considered the main target through which OEA exerts its actions, including the neuroprotective effects observed in PCD mice [14,27]. Surprisingly, a decrease in *Pparα* gene expression was observed in PCD mice (Figure 7). This result agrees with the finding of a previous work, in which a global reduction of PPARα was observed by immunolabelling in the cerebellar vermis of PCD mice from P15 [27]. Thus, these findings support that the *pcd* mutation influences the expression of elements of the endocannabinoid system within the cerebellum [27].

Regarding the effect of OEA on gene expression, the results of short-term analyses (3 and 24 h after treatment) in unharmed WT mice showed that OEA modifies the gene expression of different pro- and anti-inflammatory factors even in a non-pathological state. It has been previously demonstrated that OEA can rapidly cross the blood–brain barrier [60,61] and modify gene expression in brain structures 3 h after systemic administration [17,62]. In fact, we demonstrated that OEA promotes an overall downregulation of the gene expression of proinflammatory mediators (i.e., *Ifnγ*, *Il1β*, *Il6*, and *Tnfα*) and an upregulation of the anti-inflammatory/neuroprotective ones (i.e., *Bdfn*, *Gap43*, and *Map2*). These findings reveal for the first time that the actions of OEA on gene expression are independent of the presence of an inflammatory process or neuronal damage.

Concerning the effect of OEA on gene expression in a pathological model of neurodegeneration and neuroinflammation, we describe for the first time a dual effect of the drug over time. It is important to point out that neuroinflammatory response is a necessary process for coping with any type of injury, but can become deleterious in severe, uncontrollable, and/or long-lasting conditions, as occurs in PCD mice [40]. In this sense, when Purkinje cell degeneration has just begun [32], OEA first increases mRNA levels of most of the factors analyzed, including proinflammatory mediators. At this time point, one would expect acute neuroinflammation to be occurring to limit the degeneration to the smallest possible cerebellar area, as has been described in other models of brain damage [9]. In this scenario, OEA could enhance this phenomenon by triggering gene expression of factors involved in the acute inflammatory response. Conversely, in the long term, when a chronic neuroinflammatory state takes place [40], OEA modulates this neuroinflammation by decreasing the gene expression of the proinflammatory mediators (i.e., *Ifnγ*, *Il1β*, *Il6*, and *Tnfα;*
Figure 7). Although direct quantitative analyses of protein expression were not performed in our study, the results found for gene expression are consistent with previous works, in which OEA treatment mediated parallel effects of both gene and protein expression in other in vitro and in vivo models of neuroinflammation [17,18,20,21,22,23,25]. In parallel, OEA administration positively regulates *Bdnf* and *Map2* expression in PCD mice, as previously reported in other in vitro and in vivo experimental models [21,63,64]. BDNF and MAP2 proteins are involved in promoting neuronal survival and maintaining microtubule stability and structure, respectively. Consequently, OEA administration through its influence on the gene expression of *Bdnf* and *Map2* might potentially contribute to delaying neuronal degeneration and restoring the impaired microtubule dynamics in PCD mice, as previously proposed [27,44]. Based on our findings, it appears that OEA treatment enhances gene expression of acute neuroinflammatory factors when required at an early stage of neurodegeneration, but also promotes a less hostile and aggressive environment for PCD cerebellar neurons when the chronic inflammatory state is maintained over time.

Finally, to determine whether the aforementioned effects of OEA could be related to PPARα receptor, as previously demonstrated for the histological and behavioral neuroprotective effects in PCD mice [27], we also analyzed its gene expression after OEA treatment. We found that OEA administration increased *Pparα* gene expression even eighteen days after the treatment (Figure 7), as reported in other animal models [18,22,62]. These results also agree with previous findings in which increased cerebellar PPARα labeling was observed in PCD mice following OEA administration [27]. Together, these findings support that the anti-inflammatory and neuroprotective OEA properties are directly dependent on PPARα. In fact, no anti-inflammatory effects were observed in PPARα-silenced cells [18] and no neuroprotection was exerted in PCD mice when pretreated with the PPARα antagonist GW6471 before OEA administration [27]. However, due to the pleiotropic effect of OEA, the involvement of other receptors or an indirect effect of OEA in other cell populations cannot be ruled out.

Directly linked with neuroinflammation, microglia plays a crucial role in its regulation, responding to a changing environment to acquire specialized inflammatory or reparative phenotypes in an attempt to return the CNS to its physiological state [1,3,4,5]. In this sense, previous works have reported an exacerbated microgliosis in the cerebellum of PCD mice characterized by strong morphological alterations from branched to hypertrophied and globose cells, increased cell proliferation, changes in cell distribution, and leukocyte recruitment towards the cerebellar parenchyma [8,10,40]. The results of our quantitative analyses on microglial density and distribution in PCD mice agree with the observations of the aforementioned works [8,40]. Thus, we have shown that the density of Iba1-positive cells is higher than that of WT mice and is mainly located in the LI-IX as well as in the outermost layers of the cerebellar vermis (i.e., ML and PCL). Interestingly, these areas correspond to the cerebellar lobes where neuronal loss is more aggressive and to the layers of the cerebellar cortex where dendritic arbors and Purkinje cell somas are located, i.e., where primary degeneration takes place [32]. In contrast, a lower density of microglia was found in LX where Purkinje cells are more resistant to degeneration [32,44,45]. On the contrary, WT microglia were also observed in all cerebellar lobes and layers, although mainly located in the GL, as previously reported [40,65]. Furthermore, PCD microglial cells showed a more activated marker-based profile compared to WT as the expression of all three microglial markers analyzed (CD45, CD16/32, and CD206) increased (Figure 7). This finding also shows that both proinflammatory and anti-inflammatory microglial populations could be involved at the same time when a brain insult occurs, and that the spectrum of microglial response is more complex than the classic dichotomous classification [4,7]. There is evidence that microglial cells express a functional endocannabinoid system, and their maturation, differentiation, and activation are also under the regulatory influence of cannabinoids [48]. Our results indicate that treatment with the endocannabinoid OEA decreases the peak of Iba1-positive cells observed in PCD mice in almost all the different analyses carried out (total, per lobe, and per layer; Figure 7). In addition, OEA-treated mice show a higher percentage of neuroprotective CD206 microglial population, which agrees with previous works in which endocannabinoids modulate inflammatory responses by promoting an M2 phenotype [25,48]. Finally, regarding leukocyte recruitment, OEA treatment reduces the overall leukocyte density, which was increased in PCD mice (Figure 7; ref. [10]). This phenomenon could be related both to reduced leukocyte attraction due to a diminished cerebellar inflammatory state and to the effect of OEA on maintaining blood–brain barrier integrity, as reported previously [62,66]. Moderate leukocyte recruitment may present a protective phenotype and provide injury attenuation, whereas massive infiltration has been reported to contribute to neuroinflammation and aggravate brain injury [9]. In this sense, OEA treatment, by decreasing overall leukocyte density in the cerebellar parenchyma, could also be promoting a less neurotoxic environment that would protect surrounding neurons from degeneration in PCD mice [27]. Based on our findings, we cannot discard the direct or indirect involvement of other neuroimmune cells or even a peripheral effect of OEA, as previous studies suggested [21,67].

Our study provides evidence of the anti-inflammatory and immunomodulatory actions of OEA treatment in a model of neuronal degeneration. First, OEA modulates cerebellar gene expression by decreasing the RNA levels of proinflammatory mediators and increasing those of neuroprotective factors. Second, OEA counteracts microgliosis by decreasing microglial density and shifting their phenotype towards an anti-inflammatory and reparative state. Third, OEA treatment reduces the overall density of leukocytes in the cerebellar parenchyma, which, together with the other effects mentioned above, may be contributing to a less neurotoxic and more neuroprotective environment. Given the importance of neuroinflammation and exacerbated microgliosis in the pathophysiology, course, and prognosis of neurodegeneration, our findings provide further support for clinical trials aimed at assessing the beneficial effects of OEA as a therapeutic approach for modulating the inflammatory response in neurodegenerative diseases.

## 4. Materials and Methods

### 4.1. Animals

WT and *pcd*^1*J*^ mutant mice in a C57BL/DBA background were obtained from Jackson Laboratories (Bar Harbor, ME, USA) and employed in this study. Animals were sorted into groups according to genotype, treatment, and age at the time of analysis (*n* = 5–7 animals per experimental group; see Section 4.3). Males and females were indistinctly used in this study since no dimorphic differences have been reported for the variables considered [32]. As PCD mice are not suitable for breeding, the colony was maintained by mating heterozygous animals and genotyping their offspring as previously described [68]. Mice were housed under a 12/12 h light/dark cycle at constant room temperature and humidity and fed ad libitum with water and special rodent chow (Envigo, Indianapolis, IN, USA) at the animal facilities of the University of Salamanca (Spain). All animal protocols were approved by the Bioethics Committee of the Universidad de Salamanca (reference number #613) and were performed in compliance with the guidelines established by European (Directive 2010/63/UE) and Spanish (RD118/2021, Law 32/2007) legislation [47]. Every effort was made to ensure animal welfare, minimize animal pain and distress, and use the smallest sample size necessary to produce statistically relevant results.

### 4.2. OEA Administration

OEA ∼98% TLC (CAS: 111-58-0; Sigma-Aldrich, St. Luis, MO, USA) was freshly dissolved in 100% (*v*/*v*) ethanol (VWR Chemicals, Solon, OH, USA) and diluted in H_2_O Elix on the day it was administered to avoid drug degradation. OEA was administered at P12 by intraperitoneal (i.p.) injection at a dose of 10 mg/kg of body weight (b.w.) in a volume of 10 µL/g b.w., described as the optimal neuroprotective dose in PCD mice in previous work [27]. PCD-untreated animals (referred to “PCD” in figures) were injected i.p. with 0.9% (*w*/*v*) NaCl (Sigma-Aldrich). All animals were treated at 10:00 a.m. and their b.w. was monitored due to the anorexigenic effect of the drug, although no changes in the body weight of PCD mice were previously reported at the OEA dose administered [14,27].

### 4.3. Tissue Sample Collection and Processing

Different tissue sample collection techniques were utilized based on the purpose of the experiment carried out, as shown in Figure 1a. For the molecular analyses (*n* = 6 animals per experimental group), the modulatory effect of OEA was measured in the short- (3 and 24 h after treatment based on a previous study [62]), medium- (P20), and long term (P30; Figure 1a). For exploring the short-term effects of OEA (both after 3 and 24 h), only WT mice were used, since at this age (P12–13) PCD animals do not exhibit any differences as compared to their WT littermates [32,41,43]. That is, under short-term conditions, only standard OEA effects in an unharmed animal were evaluated. By contrast, WT, untreated PCD, and OEA-treated PCD mice were used for the rest of the analyses, allowing the putative neuroprotective effects of the drug at different time points of the neurodegenerative process to be assessed. All animals were sacrificed by decapitation and their cerebellum was collected, dissected along the sagittal midline, snap-frozen in a liquid nitrogen tank, and stored at −80 °C until used for RNA extraction. For histological analyses (*n* = 5–7 animals per experimental group), mice were deeply anesthetized at P30 (Figure 1a) and intracardially perfused with 0.9% (*w*/*v*) NaCl and 100 µL of 1000 U/mL heparin (Sigma-Aldrich) for 1 min, followed by modified Somogyi’s fixative solution containing 4% (*w/v*) paraformaldehyde (VWR Chemicals) and 15% (*v*/*v*) saturated picric acid (Probus S.A., Badalona, Spain) in 0.1 M phosphate buffer, pH 7.4 (PB) for 15 min. Brains were dissected out and kept in the same fixative solution for 2 h at room temperature. Then, they were washed with PB and cryoprotected with 30% (*w*/*v*) sucrose (VWR Chemicals) in PB overnight at 4 °C. Cerebella were cut into 30 μm thick sagittal sections using a freeze-sliding microtome (Jung SM 2000; Leica Instruments, Wetzlar, Germany). Sections were washed with phosphate-buffered saline (PBS), pH 7.4, and stored in a cryoprotective solution containing 30% (*v*/*v*) glycerol (VWR Chemicals) and 30% (*v*/*v*) ethylene glycol (Sigma-Aldrich) in 0.1 M PB at −20 °C until used for immunostaining.

### 4.4. RNA Isolation, Reverse Transcription, and Quantitative PCR Analyses

To test whether OEA treatment exerts a general immunomodulatory effect on the PCD cerebellum, changes in gene expression of different proinflammatory, anti-inflammatory, and neuroprotective factors (listed in Table 5) were analyzed with RT-qPCR. For this purpose, the left halves of cerebella were weighed, and total cytoplasmic RNA was isolated and purified using the column-based PureLink^TM^ RNA Mini Kit (Invitrogen, Thermo Fisher Scientific, Waltham, MA, USA) and PureLink^TM^ DNase Set (Invitrogen, Thermo Fisher Scientific). From each sample, 1000 ng of total RNA was converted to cDNA using the High-Capacity cDNA Reverse Transcription Kit (Applied Biosystems, Foster City, CA, USA). Quantitative changes in mRNA were measured by real RT-qPCR using the corresponding cDNA, PowerUp SYBR Green Master Mix (Applied Biosystems), and the specific pairs of primers listed in Table 1.

RT-qPCR assays were carried out with the amplification cycling conditions described in the manufacturer’s protocol on QuantStudio^TM^ 7 Flex RT PCR System (Applied Biosystems). Three replicates of each biological sample were analyzed. *Gapdh* was used as an endogenous control to normalize the data and relative gene expression was calculated as the fold change expression using the normalized Cycles to Threshold (Ct) values. Fold change is the expression ratio: if its value is >1, the gene is considered to be upregulated. By contrast, if the ratio is <1, gene expression is considered to be downregulated as compared to the control gene. Standard and melting curve analyses were performed for all primers to ensure the efficiency and specificity in the amplification of the genes of interest (Appendix A).

### 4.5. Immunofluorescence

Free-floating sections were washed with PBS (3 × 10 min) and incubated at 4 °C for 72 h with continuous rotation in medium containing 0.2% (*v*/*v*) Triton X-100 (Probus S.A.), 5% (*v*/*v*) normal donkey serum (Sigma-Aldrich), and the following primary antibodies diluted in PBS: rat anti CD16/32 (1:200; #553142, BD Biosciences-Pharmigen, Franklin Lakes, NJ, USA), goat anti-CD206 (1:200; #AF2535, R&D Systems, Minneapolis, MN, USA), rat anti-CD45 (1:1000; #MCA1388, Bio-Rad Laboratories, Hercules, CA, USA), and rabbit anti-Iba1 (1:1000; #019-19741, Wako Pure Chemical Industries, Osaka, Japan). Then, sections were incubated with the appropriate Cy2- or Cy3-conjugated secondary antibodies (1:500; Jackson ImmunoResearch, West Grove, PA, USA) in PBS for 2 h at room temperature and counterstained with DAPI (4’,6-diamidino-2-phenylindole; 1:10,000; Sigma-Aldrich) to identify cell nuclei. Sections were washed with PBS, mounted on gelatin-coated microscope slides (VRW), and covered with fresh anti-fading mounting medium (70% *v*/*v* glycerol, 5% *w*/*v* n-propyl gallate, 0.42% *w*/*v* glycine, 0.03% *w*/*v* azide). For all different combinations of antibodies used in double-immunofluorescence experiments (Iba1/CD16/32; Iba1/CD206 and Iba1/CD45), two sagittal sections of the cerebellar vermis (distance: 180 µm) of each mouse were evaluated.

### 4.6. Microscopy Visualization and Cell Counting

Immunostained cerebellar vermis sections were visualized under an Olympus Provis AX70 epifluorescence microscope (Olympus, Tokyo, Japan) equipped with an Olympus DP70 digital camera (12.5 MP, Olympus), and digital images were captured with the 10× objective from lobes III, VIa, VIII, and X of each section. All cell quantifications were performed separately for (a) the different lobes: I to IX vs. X; and (b) the different layers of the cerebellar cortex: ML + PCL vs. GL. Data from lobes III, VIa, and VIII were averaged and named as “Lobe I to IX”. These different regions of interest (ROI) were chosen based on previous studies reporting that cerebellar neurodegeneration and microgliosis in PCD mice are not homogenous across all lobes and layers of the cerebellum [10,32,40].

### 4.7. Microglial Density Quantification

Immunofluorescent Iba1-positive cells were manually counted in 24 digital images from six sections of cerebellar vermis per animal. Quantification was expressed as the mean number of Iba1-positive cells per area (mm^2^) for each ROI and experimental group. The different ROI areas were measured with ImageJ software (NIH, Bethesda, MD, USA).

### 4.8. Microglial Phenotype Characterization

For exploring the phenotype of microglia in the cerebellar vermis, three different microglial markers were employed: CD16/32 (proinflammatory marker), CD206 (anti-inflammatory/tissue repair marker), and CD45^low^ (reactive microglial marker) [48]. Double-positive Iba1 and CD-immunofluorescent cells with clear microglial morphology were manually counted in 8 digital images from two sections of the cerebellar vermis per combination of antibodies and animal. Quantification was expressed as the mean percentage of Iba1/CD double-labeled elements out of the total Iba1-positive cells for each ROI and experimental group.

### 4.9. Leukocyte Density Quantification

Purkinje cell degeneration in PCD mice triggers a specific attraction of peripheral leukocytes, increasing their infiltration into the cerebellar parenchyma [10], which might contribute to the neuroinflammatory state of the cerebellum in this animal model. To examine whether OEA treatment modifies the pattern of leukocyte density and distribution in the cerebellar parenchyma, leukocytes were quantified as the mean number of high positive CD45 round- or rod-shaped cells [10,50] per area (mm^2^). Leukocytes were clearly distinguishable from microglial cells, which are characterized by a branched or globose morphology (depending on the state) and low CD45 positive intensity [49]. Eight images from two sections of the cerebellar vermis of each animal were employed for quantifications. It is important to clarify that this quantitative analysis was performed considering all the different leukocyte populations positive for the pan-leukocyte marker CD45 [49].

### 4.10. Statistical Analyses

Data are represented both as individual points for each value and as the mean ± standard error of the mean (SEM). Homoscedasticity and normality were checked before all the statistical analyses by performing Kolmogorov–Smirnov’s and Levene’s tests. For molecular analyses, the non-parametric Kruskal–Wallis’ test was carried out for multiple comparisons (Kolmogorov–Smirnov’s and Levene’s test *p* values were < 0.05). For histological analyses, one-way ANOVA followed by Bonferroni’s post hoc test or Student’s *t*-test were employed when appropriate (Kolmogorov–Smirnov’s and Levene’s test *p* values > 0.05). A *p*-value < 0.05 was set as the minimum level of statistical significance. *p*-values for the most relevant results and simple pairwise comparisons are included in the main text, while those obtained from multiple comparisons are summarized in Table 1, Table 2, Table 3 and Table 4. All analyses and graphical representations were performed using SPSS software version 26 for Windows (IBM, Armonk, NY, USA) and GraphPad Prism software version 9.0.2 for Windows (GraphPad, San Diego, CA, USA), respectively.

## Figures and Tables

**Figure 1 ijms-24-09691-f001:**
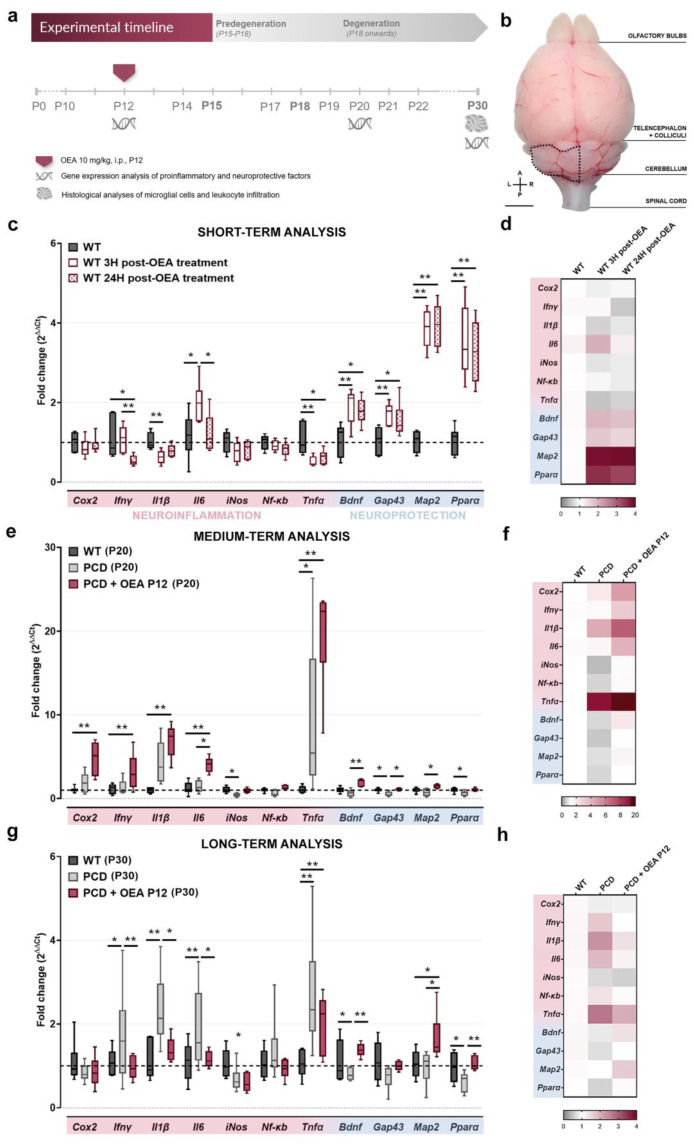
Experimental design and effect of OEA on mRNA levels of different inflammatory/neuroprotective factors in the short-, medium-, and long term. (**a**) Schematic representation of the experimental design and timeline followed in this study. (**b**) Dorsal view of the brain of a 3-week-old WT mouse; the dotted line indicates the specific cerebellar region employed for all mRNA experiments. (**c**,**d**) Graph showing relative mRNA levels of different inflammatory markers and neuroprotective factors 3 and 24 h after OEA treatment in 12-day-old WT healthy mice (**c**) and their corresponding heatmap gene expression (**d**). (**e**,**f**) Graph showing relative mRNA levels of different inflammatory markers and neuroprotective from WT, untreated PCD (PCD), and OEA-treated PCD mice (PCD + OEA) at P20 (**e**) and their corresponding heatmap gene expression (**f**). (**g**,**h**) Graph showing relative mRNA levels of different inflammatory markers and neuroprotective factors from the same experimental groups as in (**e**,**f**) at P30 (**g**) and their corresponding heatmap gene expression (**h**). Note that a striking increase in *Tnfα* gene expression was detected at the onset of the Purkinje cell degeneration process in both PCD groups, and OEA-treated mice showed an overall reduction in mRNA levels of proinflammatory markers and an increase in neuroprotective factors both in the short and the long term, even in an unharmed organism. n = six animals per experimental group (three replicates per biological sample). Kruskal–Wallis’ test for c, e, g.* *p* < 0.05; ** *p* < 0.01 (specific *p*-values for multiple comparisons in Table below). Scale bar: 2 mm. A, anterior; I.P., intraperitoneal; L, left; P, posterior; R, right. Adapted from [47].

**Figure 2 ijms-24-09691-f002:**
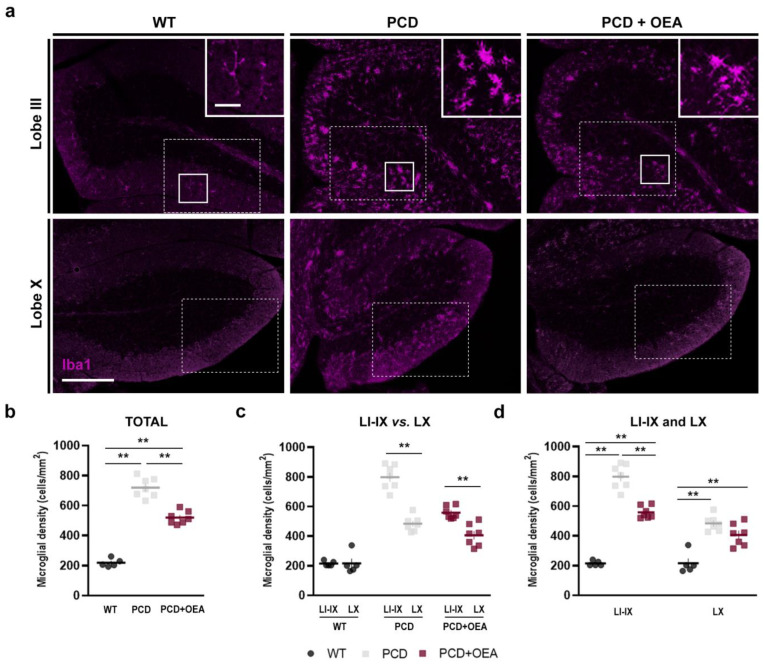
Effect of OEA on microglial density in the cerebellum (total and per lobe). (**a**) Immunolabeling for Iba1 (magenta) showing microglial cells in lobe III (as a representation of lobes I to IX) and lobe X of the cerebellar vermis of WT, PCD, and OEA-treated PCD mice at P30. Note the difference in microglial cell morphology in the zoom inset: branched in WT mice and hypertrophied/globose in PCD mice, regardless of treatment. Solid boxes show zoom inset region and dotted boxes show the position at which micrographs of Figure 3a were taken. (**b**–**d**) Quantification of the effect of OEA on total (**b**) and per lobe (**c**,**d**) microglial density; note that OEA treatment decreases the density of Iba1-positive cells in PCD mice when both whole cerebellar vermis and LI to IX are analyzed. Moreover, in contrast to WT mice, an increase in the density of Iba1-positive cells is found in LI to IX compared to LX in both PCD groups. *n* = 5–7 animals per experimental group. One-way ANOVA followed by Bonferroni’s post hoc test for (**b**,**d**); and Student’s *t* test for (**c**). ** *p* < 0.01 (specific *p*-values for multiple comparisons in Table 2). Scale bar: 100 µm, 20 µm (zoom insets). Adapted from [47].

**Figure 3 ijms-24-09691-f003:**
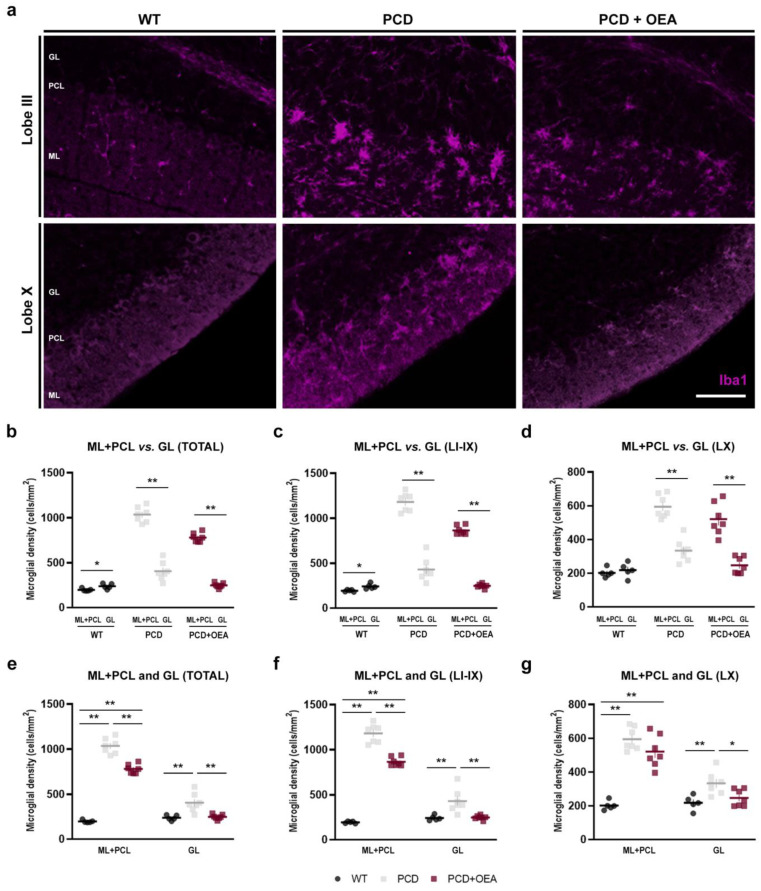
Effect of OEA on microglial density per cerebellar cortex layers. (**a**) Magnified images of the dotted boxes in Figure 2a showing Iba1-labeled microglia (magenta) in the different layers of the cerebellar cortex of lobe III (as a representation of LI-IX) and X of WT, PCD, and OEA-treated PCD mice at P30. (**b**–**d**) Quantification of microglial density in the total cerebellar cortex (**b**), lobe I to IX (**c**), and lobe X (**d**) for each experimental group, and comparison between layers; note that, unlike WT, the density of Iba1-positive cells is higher in the outermost layers (ML + PCL) compared with the deepest layer (GL) in both PCD groups for all comparisons. (**e**–**g**) Quantification of the effect of OEA on microglial density in the entire cerebellar cortex (**e**), lobe I to IX (**f**), and lobe X (**g**), per layer, and comparison among the experimental groups; note that values for OEA-treated PCD are always midway between those of WT and untreated PCD mice for most comparisons. *n* = 5–7 animals per experimental group. Student’s *t*-test for (**b**–**d**) and one-way ANOVA followed by Bonferroni’s post hoc test for (**e**–**g**). * *p* < 0.05; ** *p* < 0.01 (specific *p*-values for multiple comparisons in Table 2). Scale bar: 50 µm. GL, granular layer; ML, molecular layer; PCL, Purkinje cell layer. Adapted from [47].

**Figure 4 ijms-24-09691-f004:**
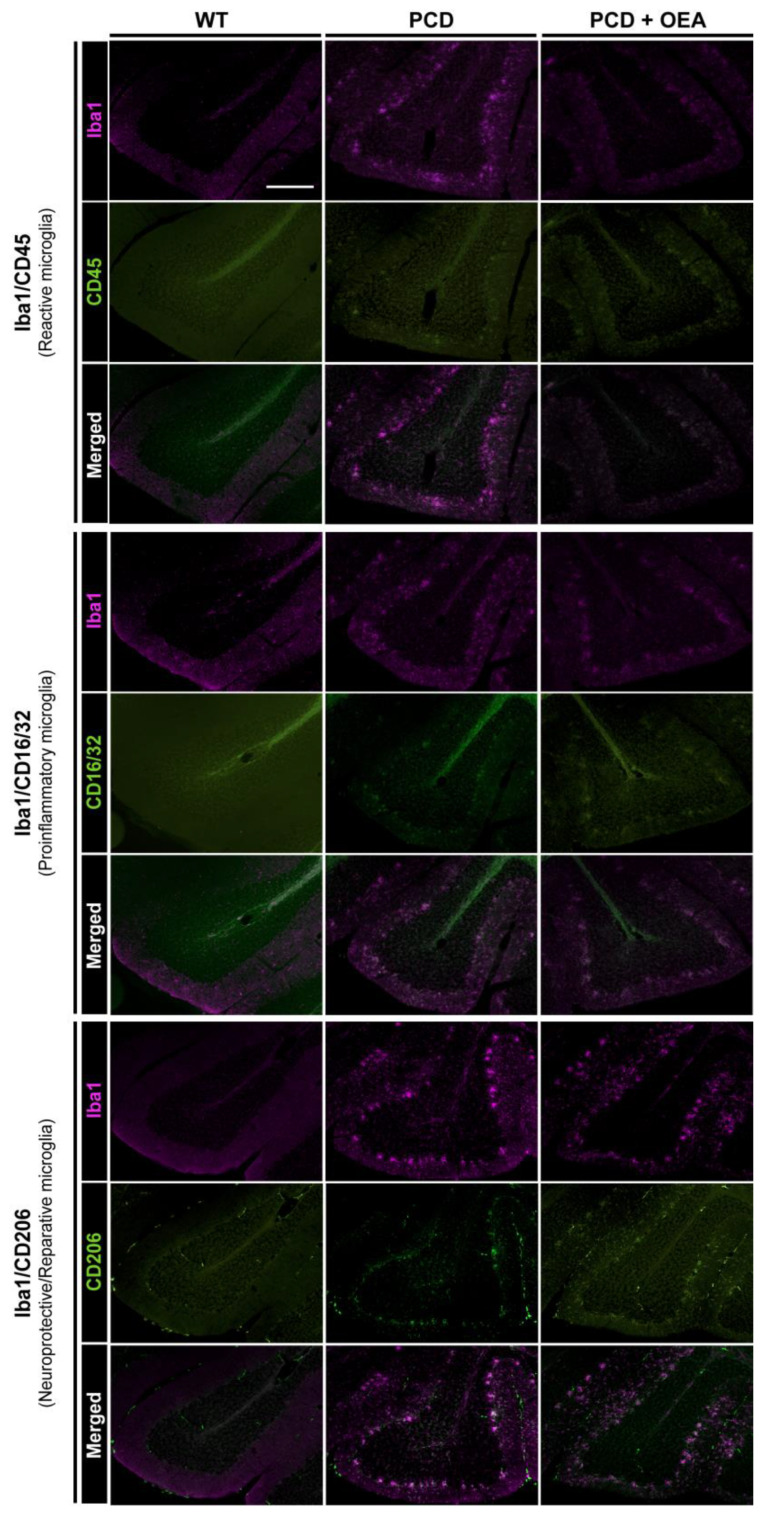
Marker-based characterization of microglia in the cerebellum of WT, PCD, and OEA-treated PCD mice at P30. At the (**top**), immunolabeling for Iba1 (magenta) and CD45 (green), i.e., reactive microglial cells. In the (**center**), immunolabeling for Iba1 (magenta) and CD16/32 (green), i.e., proinflammatory microglial cells. At the (**bottom**), immunolabeling for Iba1 (magenta) and CD206 (green), i.e., neuroprotective or tissue-reparative microglial cells. Overall, note the higher expression intensity of all the markers analyzed in PCD groups, both untreated and OEA-treated, compared to that of WT mice. Quantitative changes shown in Figure 5. Images in this figure are located in lobe VIII of the cerebellar vermis and presented at the same magnification. Scale bar: 100 µm. Adapted from [47].

**Figure 5 ijms-24-09691-f005:**
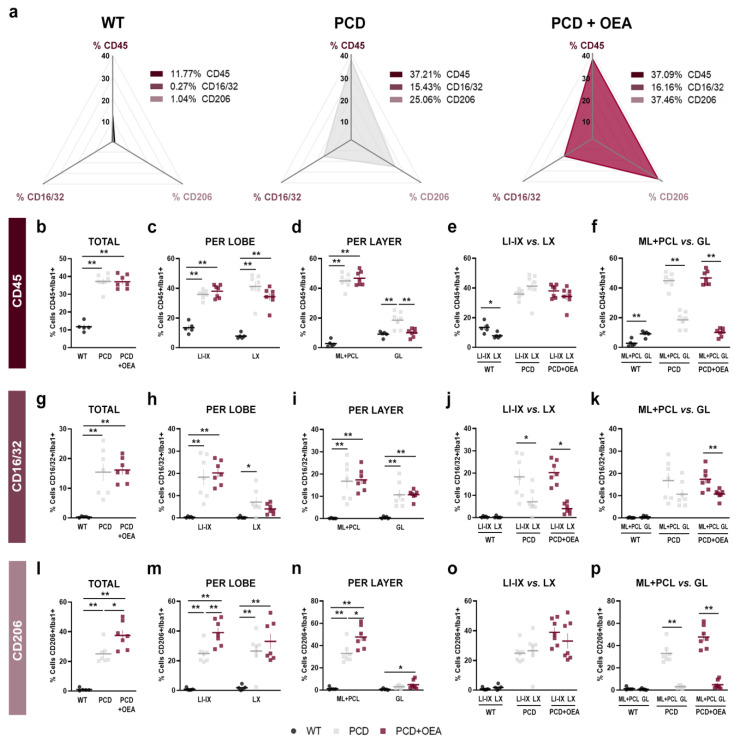
Quantitative analyses of microglial populations CD45, CD16/32, and CD206 markers in the cerebellum. (**a**) Prism representation of the percentage of CD45/Iba1, CD16/32/Iba1, and CD206/Iba1 double-positive cells in the cerebellum for each experimental group: WT, PCD, and OEA-treated PCD mice at P30. Note that WT mice show a lower percentage of all microglial populations compared to that of both PCD groups and that an increase of the anti-inflammatory CD206 marker can be observed in the OEA-treated PCD prism. (**b**–**f**) Quantification of the percentage of reactive CD45 microglia in total (**b**), per lobe (**c**,**e**), and per layer (**d**,**f**). (**g**–**k**) Quantification of the percentage of proinflammatory CD16/32 microglia in total (**g**), per lobe (**h**,**j**), and per layer (**i**,**k**). (**l**–**p**) Quantification of the percentage of anti-inflammatory CD206 microglia in total (**l**), per lobe (**m**,**o**), and per layer (**n**,**p**). Note that the most evident effect of OEA treatment is the increase in the percentage of the anti-inflammatory CD206 microglial marker for all the comparisons analyzed. *n* = 5–7 animals per experimental group. One-way ANOVA followed by Bonferroni’s post hoc test for (**b**–**d**,**g**–**i**,**l**–**n**); and Student’s *t*-test for (**e**,**f**,**j**,**k**,**o**,**p**). * *p* < 0.05; ** *p* < 0.01 (specific *p*-values for multiple comparisons in Table 3). Adapted from [47].

**Figure 6 ijms-24-09691-f006:**
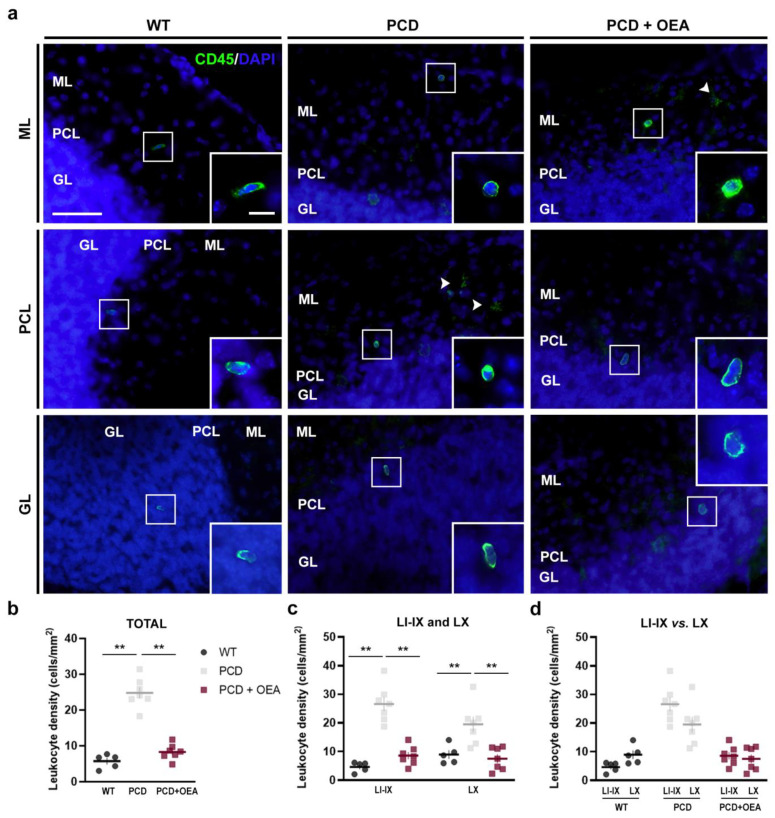
Effect of OEA on the overall leukocyte density and their distribution per lobe in the cerebellum. (**a**) Micrographs of CD45^high^-labeled leukocytes (green) with a clear rounded or rod-like morphology, distributed in the different layers of the cerebellum of WT, PCD, and OEA-treated PCD mice at P30; nuclei are stained with DAPI (blue); white arrows point out microglia with a clear distinguishable morphology. (**b**–**d**) Quantification of overall leukocyte density in the total cerebellar vermis (**b**) and per lobe (**c**,**d**). Note that the increased leukocyte density observed in untreated PCD mice was prevented in those animals treated with OEA, showing similar values to those of WT. *n* = 5–7 animals per experimental group. One-way ANOVA followed by Bonferroni’s post hoc test for (**b**,**c**); and Student’s *t*-test for (**d**). ** *p* < 0.01 (specific *p*-values for multiple comparisons in Table 4). Scale bar: 20 µm, 5 µm (zoom inset). GL, granular layer; ML, molecular layer; PCL, Purkinje cell layer. Adapted from [47].

**Figure 7 ijms-24-09691-f007:**
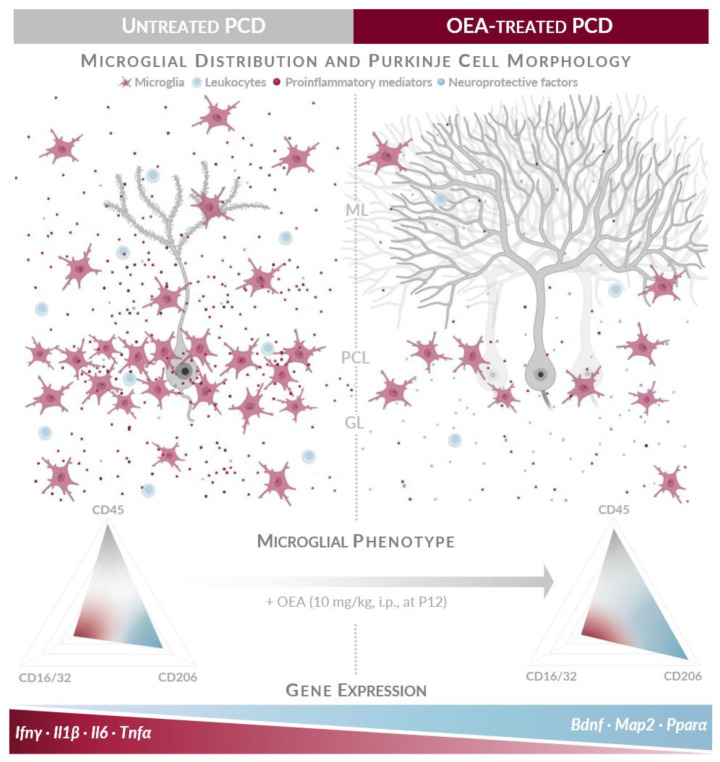
OEA treatment modulates neuroinflammation in the model of selective neuronal degeneration PCD by (top to bottom) (i) decreasing microglial density and modulating its phenotype towards an anti-inflammatory/reparative state; (ii) preventing massive leukocyte infiltration into the cerebellar parenchyma; and (iii) downregulating gene expression of proinflammatory mediators and upregulating that of neuroprotective ones. All these effects together could contribute to a less neurotoxic and more neuroprotective environment that promotes the neuroprotection of Purkinje cells demonstrated in [27]. GL, granular layer; ML, molecular layer; PCL, Purkinje cell layer. Partially created with Biorender.com. Modified from [47].

**Table 1 ijms-24-09691-t001:** Comparisons and specific *p*-values for short-, medium-, and long-term gene expression analyses.

Fold Change	Comparisons and *p*-Values		
Short Term (~P12)	WT vs. WT 3H Post-OEA	WT vs. WT 24H Post-OEA	WT 3H Post-OEA vs. WT 24H Post-OEA
*Cox2*	0.459	0.459	0.459
*Ifnγ*	0.829	**0.013**	**0.007**
*Il1β*	**0.004**	0.094	0.234
*Il6*	**0.014**	0.957	**0.020**
*iNos*	0.158	0.158	0.158
*Nfκb*	0.272	0.272	0.272
*Tnfα*	**0.003**	**0.040**	0.358
*Bdnf*	**0.009**	**0.035**	0.626
*Gap43*	**0.004**	**0.046**	0.304
*Map2*	**0.004**	**0.003**	0.914
*Pparα*	**0.002**	**0.006**	0.746
**Medium term (P20)**	WT vs. PCD	WT vs. PCD + OEA	PCD vs. PCD + OEA
*Cox2*	1.000	**0.005**	0.080
*Ifnγ*	0.097	**0.002**	0.143
*Il1β*	0.052	**0.002**	0.838
*Il6*	1.000	**0.007**	**0.015**
*iNos*	**0.028**	10.000	0.069
*Nfκb*	0.231	0.231	0.231
*Tnfα*	**0.044**	**0.002**	0.583
*Bdnf*	0.838	0.080	**0.003**
*Gap43*	**0.029**	10.000	**0.017**
*Map2*	0.913	0.281	**0.021**
*Pparα*	**0.034**	0.123	0.059
**Long term (P30)**	WT vs. PCD	WT vs. PCD + OEA	PCD vs. PCD + OEA
*Cox2*	0.587	0.587	0.587
*Ifnγ*	**0.035**	0.626	**0.009**
*Il1β*	**0.002**	0.387	**0.031**
*Il6*	**0.004**	0.589	**0.020**
*iNos*	**0.050**	**0.050**	**0.050**
*Nfκb*	0.548	0.548	0.548
*Tnfα*	**0.002**	**0.009**	0.626
*Bdnf*	**0.011**	0.787	**0.005**
*Gap43*	0.082	0.082	0.082
*Map2*	0.330	**0.015**	**0.044**
*Pparα*	**0.012**	0.787	**0.005**

Bold values indicate statistical significance at *p* ≤ 0.05. *Bdnf*, brain-derived neurotrophic factor gene; *Cox*, cyclooxygenase gene; *Gap*, growth-associated protein gene; *Gapdh*, glyceraldehyde-3-phosphate dehydrogenase gene; *Ifn*, interferon gene; *Il*, interleukin gene; *iNos*, inducible nitric oxide synthase gene; *Map*, microtubule-associated protein gene; *Nf*, nuclear factor gene; *Tnf*, tumor necrosis factor gene.

**Table 2 ijms-24-09691-t002:** Comparisons and specific *p*-values for microglial density analysis.

Microglia Density	Comparisons and *p*-Values		
	WT vs. PCD	WT vs. PCD + OEA	PCD vs. PCD + OEA
Total	**<0.001**	**<0.001**	**<0.001**
LI-IX	**<0.001**	**<0.001**	**<0.001**
LX	**<0.001**	**<0.001**	0.121
ML + PCL (Total)	**<0.001**	**<0.001**	**<0.001**
ML + PCL (LI-IX)	**<0.001**	**<0.001**	**<0.001**
ML + PCL (LX)	**<0.001**	**<0.001**	0.220
GL (Total)	**0.002**	1.000	**0.001**
GL (LI-IX)	**0.004**	1.000	**0.002**
GL (LX)	**0.008**	1.000	**0.029**

Bold values indicate statistical significance at *p* < 0.05. GL, granular layer; L, lobe; ML, molecular layer; PCL, Purkinje cell layer.

**Table 3 ijms-24-09691-t003:** Comparisons and specific *p*-values for microglial phenotype analysis.

Microglia Percentage	Comparisons and *p*-Values		
	WT vs. PCD	WT vs. PCD + OEA	PCD vs. PCD + OEA
CD45 (Total)	**<0.001**	**<0.001**	1.000
CD45 (LI-IX)	**<0.001**	**<0.001**	0.875
CD45 (LX)	**<0.001**	**<0.001**	0.218
CD45 (ML + PCL)	**<0.001**	**<0.001**	1.000
CD45 (GL)	**0.002**	1.000	**0.003**
CD16/32 (Total)	**0.001**	**<0.001**	1.000
CD16/32 (LI-IX)	**0.001**	**<0.001**	1.000
CD16/32 (LX)	**0.016**	0.286	0.402
CD16/32 (ML + PCL)	**0.001**	**0.001**	1.000
CD16/32 (GL)	**0.001**	**0.001**	1.000
CD206 (Total)	**<0.001**	**<0.001**	**0.011**
CD206 (LI-IX)	**<0.001**	**<0.001**	**0.003**
CD206 (LX)	**0.005**	**0.001**	0.859
CD206 (ML + PCL)	**<0.001**	**<0.001**	**0.011**
CD206 (GL)	0.431	**0.044**	0.619

Bold values indicate statistical significance at *p* < 0.05. GL, granular layer; L, lobe; ML, molecular layer; PCL, Purkinje cell layer.

**Table 4 ijms-24-09691-t004:** Comparisons and specific *p*-values for overall leukocyte density analysis.

Leukocyte Density	Comparisons and *p*-Values		
	WT vs. PCD	WT vs. PCD + OEA	PCD vs. PCD + OEA
Total	**<0.001**	0.504	**<0.001**
LI-IX	**<0.001**	0.477	**<0.001**
LX	**0.009**	1.000	**0.002**

Bold values indicate statistical significance at *p* < 0.05. L, lobe.

**Table 5 ijms-24-09691-t005:** List of RT-qPCR primer sequence details.

Gen Name ^1^	Forward Primer (3′-5′)	Reverse Primer (5′-3′)
*Bdnf*	GTGGTGTAAGCCGCAAAGA	AACCATAGTAAGGAAAAGGATGGTC
*Cox2*	GGTCATTGGTGGAGAGGTGTA	TGAGTCTGCTGGTTTGGAATAG
*Gap43*	GCTGGTGCATCACCCTTCT	TGGTGTCAAGCCGGAAGATAA
*Gapdh*	GCCTATGTGGCCTCCAAGGA	GTGTTGGGTGCCCCTAGTTG
*Ifnγ*	CAGCAACAGCAAGGCGAAAAAGG	TTTCCGCTTCCTGAGGCTGGAT
*Il1β*	TGCTCATGTCCTCATCCTGGAAGG	TCGCAGCAGCACATCAACAAGAG
*Il6*	GAGGATACCACTCCCAACAGACC	AAGTGCATCATCGTTGTTCATACA
*iNos*	CTTTGCCACGGACGAGAC	AACTTCCAGTCATTGTACTCTGAGG
*M* *ap2*	GCTGTAGCAGTCCTGAAAGGTG	CTTCCTCCACTGTGGCTGTTTG
*Nfκb*	GCTGCCAAAGAAGGACACGACA	GGCAGGCTATTGCTCATCACAG
*Pparα*	ATGCCAGTACTGCCGTTTTC	TTGCCCAGAGATTTGAGGTC
*Tnfα*	GCTTGTCACTCGAATTTTGAGA	ATGTCTCAGCCTCTTCTCATTC

^1^ Abbreviations: *Bdnf*, brain-derived neurotrophic factor; *Cox*, cyclooxygenase; *Gap*, growth-associated protein; *Gapdh*, glyceraldehyde-3-phosphate dehydrogenase; *Ifn*, interferon; *Il*, interleukin; *iNos*, inducible nitric oxide synthase; *Map*, microtubule-associated protein; *Nf*, nuclear factor; *Tnf*, tumor necrosis factor.

## Data Availability

Data supporting the findings and conclusions of this study are available from the corresponding authors upon request.

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
