# Peer review of "Oleoylethanolamide Treatment Modulates Both Neuroinflammation and Microgliosis, and Prevents Massive Leukocyte Infiltration to the Cerebellum in a Mouse Model of Neuronal Degeneration"

_ijms, 2023, doi:10.3390/ijms24119691_

Round 1

Reviewer 1 Report

In the article, the authors use a model of cerebellar degeneration (PCD), specifically of Purkinje cells, to analyse the effect of treatment with the PPARalpha receptor agonist, OEA in neuroinflammation. For this purpose, they use measurements of different anti-inflammatory and pro-inflammatory markers, as well as histochemical techniques. The topic is interesting, but the article has serious shortcomings. One of them is that the authors have decided to include all statistical data as supplementary material and not in the text, as they indicate to facilitate reading. However, the statistical data should be included in the body of the article, at least the most relevant in the text or as a table. It is unreasonable to have to refer to an appendix document for each of the results shown.

No behavioural tests are shown to support the results, so it would be necessary to include cognitive tests such as the object recognition test or some specific tests of cerebellar activity such as the balance beam, among others. The immunohistochemical images are of very poor quality and with such a wide magnification that the details of the cells cannot be seen. Therefore, an improvement is needed at this level. The authors should also justify why they use different cerebellar regions in various figures. The study of leukocyte infiltration is not very precise, and the images are again of very poor quality and detail and rule out the intervention of other populations from the periphery.

With regard to effects on gene expression, these data would need to be corroborated by protein or FACS analysis.  

In Figure 7, the authors indicate a possible explanation for the modulation exerted by OAS in PCD, but the data they indicate on the microglial phenotype are confusing, a review of the percentage data provided is needed.

The role of astrocytes in neuroinflammation has not been taken into account in this study, but considering the importance of this cell type in the cerebellum, it is something to be taken into account. Furthermore, previous data in ischaemia models show that OAS administration inhibits glial activation by modulating astrocyte activation.

In order to be able to affirm that the observed effects are mediated by PPARalpha activation, it is important to be able to rely on pharmacological studies using receptor antagonists.

Reviewer 2 Report

In general:

The authors' work reports how oleoylethanolamide (OEA), a PPARα agonist, can minimize the inflammatory effects and modulate the immune system of animals with Purkinje cell degeneration (PCD).

I carried out a survey of some points that I consider important to further improve the work of the authors:

- As the objective of the work is to evaluate the ability of OEA to minimize the effects of the neurodegenerative disease, it is not clear the importance of establishing a comparison with the WT group and not only with the PCD group.

- I see that the statistical analysis is described in the methodology, however I suggest the author add in the figures which test was used for the results.

- Figure 1C: I suggest changing the color of the 3h group to be standardized according to the other graphs and changing the “Fold Change (2ddct)” axis to “Fold Change (2ΔΔct)”.

- Figure 7. A figure that objectively depicts all the authors' work. Definitely something that catches the reader's attention. However, some clarification would be interesting. Should microglial phenotype values be close to 100%?

- Line 183-205. This entire paragraph is discussing a figure that was considered supplemental (figure S1) by the authors. However, if you decide to persist with this paragraph, I suggest that this figure be part of the pricnipal text or that this paragraph be more succinct. In this same line of reasoning, Figure 4 is only mentioned in line 293. It seems that it has not been explored as well as it could have. I like further explaining this image or transferring it to supplement.

- OEA affects the regulation of genes ranging from neuroinflammatory factors to neuroprotective factors. However, to affect these parameters have been detected in materials from the cerebellum. Since OEA may have induced effects directly in the CNS, is it able to cross the blood-brain barrier? 

- It is not clear what the criteria are for choosing the day to start treatment (P12) and the days of analysis (3h, 24h P20, P30). Why are the histological analyzes performed only with animals on day P30 and not with the others? 

-I would like the authors to rephrase the sentence in line 503-506 “BDNF 503 and MAP2 are involved in neuronal survival and microtubule stability and structure and 504 may also contribute to preventing neuronal degeneration and restoring impaired micro-505 tube dynamics in PCD mice [26, 43].”

Round 2

Reviewer 1 Report

The authors have satisfactorily answered all the doubts raised and considerably improved the article. Data provides that makes it much more understandable and the quality of the images improved a lot. Congratulations, on the excellent work on the archive in answering the questions raised. With this in mind, I think the article is now recommended for publication.